# Manipulating Macrophage/Microglia Polarization to Treat Glioblastoma or Multiple Sclerosis

**DOI:** 10.3390/pharmaceutics14020344

**Published:** 2022-02-01

**Authors:** Thomas Kuntzel, Dominique Bagnard

**Affiliations:** 1UMR7242 Biotechnology and Cell Signaling, Centre National de la Recherche Scientifique, Strasbourg Drug Discovery and Development Institute (IMS), University of Strasbourg, 67400 Illkirch-Graffenstaden, France; t.kuntzel@unistra.fr; 2Ecole Supérieure de Biotechnologie de Strasbourg, 67400 Illkirch-Graffenstaden, France

**Keywords:** macrophage, microglia, glioblastoma, multiple sclerosis, polarization, treatments, therapeutic

## Abstract

Macrophages and microglia are implicated in several diseases with divergent roles in physiopathology. This discrepancy can be explained by their capacity to endorse different polarization states. Theoretical extremes of these states are called M1 and M2. M1 are pro-inflammatory, microbicidal, and cytotoxic whereas M2 are anti-inflammatory, immunoregulatory cells in favor of tumor progression. In pathological states, these polarizations are dysregulated, thus restoring phenotypes could be an interesting treatment approach against diseases. In this review, we will focus on compounds targeting macrophages and microglia polarization in two very distinctive pathologies: multiple sclerosis and glioblastoma. Multiple sclerosis is an inflammatory disease characterized by demyelination and axon degradation. In this case, macrophages and microglia endorse a M1-like phenotype inducing inflammation. Promoting the opposite M2-like polarization could be an interesting treatment strategy. Glioblastoma is a brain tumor in which macrophages and microglia facilitate tumor progression, spreading, and angiogenesis. They are part of the tumor associated macrophages displaying an anti-inflammatory phenotype, thereby inhibiting anti-tumoral immunity. Re-activating them could be a method to limit and reduce tumor progression. These two pathologies will be used to exemplify that targeting the polarization of macrophages and microglia is a promising approach with a broad spectrum of applications deserving more attention.

## 1. Introduction

Macrophages are an important cellular component of the innate immune system and more precisely, a part of the mononuclear phagocytic system. They differentiate from monocytes after entering the tissues. Depending on the tissue, they will have different names such as microglial cells corresponding to the resident macrophages in the central nervous system (CNS). The macrophages can be resident or motile [1]. Concerning the origin of these cells, the first point of view is that all macrophages originate from the circulating monocytes, meaning that they derive from hematopoiesis. However, the newest consensus is that macrophages can directly originate from embryonic precursor cells that have colonized developing tissues before birth [2]. One example is the microglia, which come from the yolk sac [1]. The different populations of macrophages and their origins are presented in Figure 1. Regarding the roles of the macrophages, they perform phagocytosis, but they also have the function of antigen presenting cells (APCs) and they produce and secrete cytokines. Macrophages perform several roles after activation. There are different forms of activation corresponding to the different states of polarization: the classically activated macrophages or M1 macrophages, and the alternatively activated macrophages or M2 macrophages. Each state of polarization has different implications in the roles the macrophages exert. In M1 polarization, they mainly have pro-inflammatory and antitumor effects whereas in M2 polarization, the macrophages essentially have anti-inflammatory functions [2,3]. These opposite functions have different implications in several diseases, and they constitute potential therapeutic targets. Among them, glioblastoma and multiple sclerosis represent two examples to discuss to what extent manipulating macrophage/microglia polarization may represent an interesting therapeutic approach.

Glioblastoma (GBM) is a malignant brain tumor with a very low survival rate (one year survival ~35%) [4]. It is characterized by a high capacity of invasion, allowing the tumor to infiltrate nearby tissues. The standard of care treatment consists of surgical resection, if possible, combined with radiotherapy and temozolomide, an alkylating agent, but the prognosis remains poor [4]. Bevacizumab, an anti-VEGF antibody, initially seen as an interesting alternative showed limited efficacy and tolerance. New therapeutic possibilities are necessary to increase the patient’s survival rate and quality of life [4]. One approach is to target the tumor associated macrophages (TAMs). TAMs are peripheral macrophages or brain-intrinsic microglial cells recruited by the tumor to support its survival, growth, and migration. Interestingly, these macrophages are mostly in M2 polarization. This means that it is possible to rebalance the polarization state toward M1 in order to obtain pro-inflammatory and anti-tumor macrophages [5]. However, in inflammatory pathologies such as multiple sclerosis (MS), having only M1 polarized macrophages is associated with a deterioration of the disease. MS is the most frequent autoimmune chronic inflammatory disease of the CNS. It is characterized by the appearance of lesions where demyelination, inflammation, and glial reaction are observed. The classical course of the pathology, also called relapsing-remitting multiple sclerosis (RRMS), is punctuated by episodes of neurologic disability called relapses, which are fully or partially reversible. After 10–20 years, the disease becomes progressive, without recovery periods. About 15% of the patients have a progressive course from onset. Although there are many disease modifying medications that reduce the frequency and severity of relapse, there is no cure for the disease and no medication that prevents or reverses the progressive state [6]. An approach targeting the macrophages/microglia is also relevant for this disease. In fact, the M2 polarized cells are involved in remyelination processes, however, in MS, the macrophages/microglia are mostly in M1 polarization in favor of inflammation [3]. Some medications already on the market target the polarization state, but more are needed to expect effects on the progressive state of the pathology.

Thus, the aim of this review is to discuss the different therapeutic possibilities for the balancing of the polarization of macrophages and microglial cells. The focus will be the treatment of glioblastoma and multiple sclerosis being two archetypic diseases for which this option is becoming realistic. Some of these therapeutic agents are already on the market whereas others are in preclinical or clinical development.

## 2. The Different Polarization States of Macrophages

The concept of different macrophage activation states has existed since the nineties and in 2000, Mills et al. first termed these activation states as M1 and M2 [7]. Nonetheless, they immediately showed that assuming there are only two types of polarization is an oversimplification [7]. Indeed, M1 and M2 are the extremities of a continuum of phenotypes and possess an important plasticity. M1 polarization corresponds to a pro-inflammatory activity while M2 polarization reflects an anti-inflammatory role. There is a balance between both extremities and an imbalance can lead to pathological states. Mantovani et al. further described this plasticity by showing that the polarized macrophages differed in terms of receptor expression, effector function, and cytokine/chemokine production, depending on which extremity of the continuum the macrophages are in [8]. The polarization is also a dynamic process in so far as it can occur at any point during the inflammation, depending on the tissue microenvironment [9]. The polarization of macrophages can be regulated through three different ways: extrinsic pathways, intrinsic pathways, and the tissue microenvironment. The extrinsic pathways are the methods used in vitro with cultured macrophages by stimulating them with M1 or M2 polarizing agents. Intrinsic pathways and the tissue microenvironment are all agents secreted by the body, which influence the type of macrophage population and their polarization. Depending on all these agents, the macrophages will be more or less in a M1-like or a M2-like polarized state [9]. The polarization states of macrophages are summarized in Figure 2.

### 2.1. The M1 Polarization

Classical activation of macrophages corresponds to M1 polarization. When the macrophages shift toward this polarization, they have an enhanced capacity of secretion of pro-inflammatory cytokines such as TNF-α, IL-15, IL-23, and IL-1β, their antigen presenting capacity is amplified, their production of reactive nitrogen and oxygen intermediates is augmented and their release of IL-12—in favor of Th1 polarization of CD4+ cells—is also increased. Thus, M1-like macrophages are effector cells, exerting microbicidal and cytotoxic activities [10].

The induction of polarization depends on specific stimuli and for M1-like induction, three major stimuli are described: IFN-γ (interferon-γ), pathogens, and GM-CSF (granulocyte macrophage colony-stimulating factor). IFN-γ is the main M1 stimuli, produced by Th1 cells, NK cells, and macrophages. IFN-γ activates, through its receptor IFNGR, STAT1, which induces gene expression for cytokine receptors (e.g., IL15Rα, IL6R), cell activation markers (e.g., CD36, CD38, CD69, and CD97), and cell adhesion molecules (e.g., intercellular adhesion molecule 1 (ICAM1), integrin alpha L (ITGAL), and mucin 1). Pathogens induce the “innate” activation of macrophages through pattern recognition receptors (PRRs). The best-studied example is lipopolysaccharide (LPS), found at the bacterial outer membrane of Gram-negative bacteria, which is recognized by Toll-like receptor 4 (TLR-4). This activation causes the secretion of pro-inflammatory cytokines (e.g., IFN-β, TNF, IL-6, IL-12, and IL-1β), chemokines (e.g., CCL2, CXCL10), and antigen presentation molecules (e.g., MHC members and co-stimulatory molecules). Finally GM-CSF, produced among others by parenchyma cells and macrophages, enhances, after recognition by its receptor, antigen presentation, phagocytosis, microbicidal capacity, leukocyte chemotaxis and adhesion, and cytokine production (e.g., TNF, IL-6, IL-1β, G-CSF, and M-CSF) [11].

### 2.2. The M2 Polarization

Alternatively activated macrophages, or M2 macrophages, correspond to the other extremity of the polarization continuum. These macrophages exert different and sometimes opposite functions such as M1-like macrophages. Indeed, M2-like macrophages prevent the expansion of parasites, perform remodeling and repair of damaged tissue, have immunoregulatory effects, control inflammation, and are in favor of tumor progression by inhibiting the anti-tumoral immunity, promoting tumor survival, and facilitating angiogenesis [1,10,12]. However, the vision of one M2 polarization is oversimplified. Mantovani et al. already showed in 2004 that there were at least three different M2-like states: M2a, M2b, and M2c [13]. M2a, also called alternatively activated macrophages, are induced by IL-4 and IL-13, which are secreted by mast cells, Th2 cells, and basophiles. These cytokines activate STAT6 through the IL-4 receptor. M2a macrophages are also induced by helminth and fungal infections. The resulting effects observed are a decrease in the secretion of pro-inflammatory cytokines (e.g., TNF-α, IFN-γ, IL-6, IL-12, and IL-1β) and the expression of proteins involved in tissue repair and fibrogenesis [10,14]. M2b or type II macrophages are activated by immune complexes (ICs), TLR agonists such as LPS or IL-1R agonists. Their activation leads to the production of pro-inflammatory cytokines (TNF-α, IL-6, IL-1β) and anti-inflammatory cytokines (IL-10) and to the loss of IL-12 secretion. They have roles in Th2 activation, in upregulating antigen presentation, and in immunoregulation [11,15]. M2c macrophages, also called deactivated macrophages, are induced by IL-10, glucocorticoids, and TGF-β. They possess strong anti-inflammatory properties and promote phagocytosis of apoptotic cells [15]. A fourth state of polarization has been described in vitro: M2d macrophages, induced by TLR agonists through the adenosine receptor, lead to the loss of secretion of pro-inflammatory cytokines, the induction of the production of anti-inflammatory cytokines and vascular endothelial growth factor (VEGF). Hence, these macrophages have proangiogenic properties and exhibit attributes of TAMs [15].

### 2.3. The New Concepts about “Polarization”

In these last years, the concept of M1/M2 polarization has become more and more controversial. The concept fits well with the situation in vitro, but these models are unable to mimic the profiles observed in vivo for pathological situations. Moreover, macrophages have the capacity to develop mixed M1/M2 phenotypes, and even if there are differences between both polarizations, there are also similarities and both populations often coexist. This means that rather than focusing only on the study of entire populations of cells, a single-cell analysis could be more relevant [11,16,17]. The M1/M2 model is not adapted for the analysis of macrophages coming from specific tissues and/or from pathological conditions. Single cell RNA sequencing of macrophages shows the coexistence of many different signatures. These signatures correspond to different tissue- and disease-specific activation states and different times of activation process. Quantifying the expression for each cell of a large subset of genes associated with M1-like or M2-like states and a large subset of genes associated with early or late differentiation, is the possibility to refine the activation state of macrophages [17,18]. Notwithstanding, the single-cell RNA sequencing is more in favor of the point of view that “every macrophage is unique” [19]. Finally, it seems to be more interesting to classify the macrophages according to their function: pro-inflammatory or anti-inflammatory and for host defense, wound healing, or immune regulation [20].

The macrophages, through these functions, are of special interest for plenty of diseases and they constitute an interesting target for new therapeutics. Therefore, we will focus on two pathologies of the CNS where macrophages/microglia have different and somehow opposite roles: multiple sclerosis and glioblastoma. In MS, an exacerbation of inflammation is responsible for the lesions whereas in GBM, a hindrance of the anti-tumoral inflammatory mechanisms favors tumoral proliferation and spreading. In MS, macrophages/microglia, polarized in a M1-like phenotype have deleterious effects on the myelin sheath (Figure 3A). In GBM, polarized in a M2-like phenotype, they help the tumor cells to proliferate (Figure 3B). Re-educating these macrophages/microglia could be an interesting therapeutic option, and finally, targeting the same cells, but in an opposing way is possible for two antipodal diseases.

## 3. Macrophages/Microglia in MS

### 3.1. The M1-M2 Balance Consequences on Multiple Sclerosis

Microglia and macrophages are implicated in the pathophysiology of multiple sclerosis. On one hand, they lead to the destruction of the myelin sheath and the damage of axons by secreting pro-inflammatory cytokines, presenting antigens, and inducing oxidative stress; on the other hand, they promote remyelination, tissue repair, resolve inflammation, and perform phagocytosis of myelin debris. These contrasting effects are thought to correspond to the two major polarization states: M1-like and M2-like, respectively [3,21,22]. In fact, as discussed above, in vivo, the situation is much more complicated: the cells are able to express M1 markers as well as M2 markers simultaneously. The polarization state depends a lot on environmental signals, the length, and the combination of stimuli. It shows once more the plasticity of macrophages and microglial cells, changing from a pro-inflammatory polarization state during the acute phase of the disease, to an anti-inflammatory state during the recovery phase, but also the capacity to stay in an intermediary state between both phenotypes [3,16,22]. Interestingly, in chronic active lesions, M2 markers are lacking: they disappear when an active lesion evolves from an acute to a progressive one [16]. Targeting the imbalance between pro-inflammatory and anti-inflammatory functions of macrophages/microglia is a possible treatment option to restore polarization in favor of remyelination.

### 3.2. Treatments Influencing Polarization for MS Care Already on the Market

An important number of treatments indicated for MS care already exist and are on the market. Among these, some have been shown to indirectly target macrophage and microglia polarization. They are part of two different groups of compounds. The first group is composed of proteins and peptides including interferon β and glatiramer acetate. The second group consists of small molecules including dimethylfumarate, fingolimod/siponimod, and glucocorticoids.

#### 3.2.1. Proteins/Peptides

Interferon β is a first line background treatment for multiple sclerosis with diverse mechanisms of action. It is an anti-inflammatory cytokine that acts by increasing the production of other anti-inflammatory cytokines such as IL-10 and IL-4 by decreasing the production of pro-inflammatory cytokines such as IL-12 and IL-17 by limiting the leukocyte migration across the blood–brain barrier (BBB) and by promoting the production of nerve growth factor (NGF) in favor of neuronal survival and repair [23]. Concerning macrophages and microglia, the modifications of the expression of pro-inflammatory and anti-inflammatory cytokines directly affect their polarization state toward a M2-like phenotype. In MS patients treated with IFN-β, it was observed that the antigen presenting ability and the migration capacity of these cells was diminished and that inhibitory immune checkpoints (in particular B7-H1) were activated [24].

Glatiramer acetate is also a first line background treatment for relapsing-remitting multiple sclerosis. It is a random basic copolymer of four natural occurring amino acids: glutamic acid, lysine, alanine, and tyrosine. Its mechanism of action involves the generation of Th2 and Treg regulatory and anti-inflammatory lymphocytes, but also the induction of regulatory, anti-inflammatory, M2-like macrophages, and microglia [24]. These polarization shifts are accompanied by a diminished secretion of pro-inflammatory cytokines (e.g., TNFα and IL-12) and an augmented secretion of anti-inflammatory cytokines (IL-10 and TGF-β). The drug is also able to induce the phagocytic activity of macrophages and microglia in rats and humans, which allows for the clearance of myelin debris in favor of remyelination [24,25]. Through the APC role of macrophages, it is hypothesized that they are responsible for the amplification of Th2 cells, meaning that for this drug, macrophages and their anti-inflammatory polarization have a central duty [26].

#### 3.2.2. Small Molecules

Dimethylfumarate is also indicated for the background treatment of relapsing-remittent multiple sclerosis. Originally a treatment for psoriasis through the induction of a shift from a Th1 response to a Th2 response, the drug also induces the expression of the anti-inflammatory cytokine IL-10 by lymphocytes, but also by macrophages and microglia. Moreover, it inhibits the expression of pro-inflammatory cytokines (e.g., TNFα, IL1-β, and IL-6) by macrophages and microglia, thus promoting a M2-like phenotype [24,25,27,28].

Fingolimod is a modulator of the sphingosine-1-phosphate receptors indicated in the second line for background treatment of very active forms of multiple sclerosis. Fingolimod treatment leads to the inhibition of the egress of lymphocytes from lymphoid tissues into the circulation, thus preventing their trafficking toward the CNS. Interestingly, macrophages and microglia also express this receptor. It has been shown in human and animal models that fingolimod is able to prevent the activation of an inflammatory phenotype of these cells [24,25,29]. Fingolimod is able to modulate macrophage and microglia activation by lowering their pro-inflammatory cytokine production (e.g., TNFα), thus protecting oligodendrocytes from death and favoring the remyelination process. This mechanism corresponds to a changing of the polarization of macrophages and microglia toward a M2-like phenotype [29,30,31]. Another modulator of the sphingosine-1-phosphate receptors is the fingolimod-derived compound siponimod. Siponimod is selective for S1P1 and S1P5 receptors, and lowers the risk of adverse cardiac events mediated through the S1P3 receptor subtype [32]. This molecule protects against neurodegeneration by limiting inflammation and the recruitment of immune cells and by preventing neuronal loss. This neuroprotection is partly mediated through the interaction of siponimod with microglia, leading to an inhibition of the secretion of pro-inflammatory cytokines (IL-6 and CCL5), thereby re-educating them toward an anti-inflammatory phenotype [32,33,34]. 

Glucocorticoids are indicated for the treatment of acute relapses of MS, particularly high doses of methylprednisolone. These drugs target glucocorticoid receptors and exert their activity mainly by inhibiting T cell activation and promoting their apoptosis, but also by inhibiting the secretion of pro-inflammatory cytokines and by improving the integrity of the BBB, thus preventing the infiltration of immune cells. These effects also affect monocytes and macrophages by preventing their secretion of pro-inflammatory cytokines (like TNF-α, IL-6 and IFN-γ) and by promoting an anti-inflammatory M2-like polarization while increasing their chemotaxis. This is in favor of the resolution of the inflammation, but also for reparation and remyelination. It was noted that these effects were observed in vitro on monocytes from MS patients treated with glucocorticoids [24,35].

### 3.3. Compounds Influencing Polarization for MS Care: New Perspectives

All the drugs presented above are already on the market and have an authorization for the treatment of multiple sclerosis. Most of the drugs indicated for this pathology show effects on the polarization of macrophages and/or microglia. However, therapeutics specifically targeting these cells are lacking despite their important role in pathophysiology. We now present some drugs under development targeting macrophages, microglia, and their polarization. Two major groups of compounds are being investigated: small molecules with a synthetic origin and natural occurring compounds directly extracted from their natural source. All these compounds are immunomodulators without a unique molecular target.

#### 3.3.1. Synthetic Small Molecules

Lenalidomide is an FDA approved drug derived from thalidomide for the treatment of myelodysplastic syndromes and multiple myeloma. However, it has been shown in vitro that lenalidomide possesses repolarizing effects on macrophages. In fact, the drug can skew macrophages toward a M2-like phenotype. This was observed because an upregulation of M2 markers (Arg1, Mrc1) and an increase of the secretion of anti-inflammatory cytokines (IL-4, IL10, IL-13 and TGF-β) could be detected. Interestingly, lenalidomide does not inhibit the LPS-induced M1-like polarization, meaning that the macrophages express both M1 and M2 markers. Nonetheless, lenalidomide was also tested in vivo on the mice model of MS: Experimental Autoimmune Encephalomyelitis (EAE), obtained after exposure of the animals to myelin antigens (here a MOG-EAE model). Results show that the effect on macrophages is enough to alter their capacity to activate autoimmune CD4 T cells and to alleviate the symptoms of the disease [36].

Ethyl pyruvate is an analogue of the EMA- and FDA-approved drug dimethylfumarate. Its effect on macrophages/microglia is an inhibition of their secretion of pro-inflammatory cytokines (TNF-α, IL-6) and an inhibition of their antigen presentation capacity [37,38]. 

Another example is the immunomodulatory molecule laquinimod. The activity of this compound is mediated through the promotion of regulatory T cells, but also the decrease in activation of microglia and the inhibition of the recruitment of macrophages in the CNS [39,40,41]. Interestingly, laquinimod has already been tested in humans in phase II as well as phase III studies, which showed effects in reducing brain atrophy and limiting disability worsening. Moreover, the orally administered compound seems to be safe and well tolerated [42,43]. 

A last example is minocycline, an antibiotic of the tetracycline family, which shows neuroprotective effects in RRMS [44]. Its activity is mediated by the inhibition of T-cell migration and activation, but also by the inhibition of the proliferation and M1-like activation of macrophages and microglia by limiting the secretion of pro-inflammatory cytokines (TNF-α) and promoting the secretion of the anti-inflammatory ones (IL-10). An attenuation of the clinical course of EAE has been observed [45,46,47]. Minocycline has also been tested for its efficacy on humans and it seems to lower the risk of conversion from a clinically isolated syndrome to MS at six months but not at 24 months [48,49]. Another clinical trial assessing the effect of the combination of minocycline and interferon β-1a for RRMS showed no beneficial effect [50]. More studies are needed to better understand the mechanism of action of minocycline.

#### 3.3.2. Natural Occurring Compounds

An important number of natural occurring molecules have been tested for their repercussions on macrophage and/or microglia polarization. These have been reviewed extensively in [51,52,53]. Most of these compounds do not target only macrophages/microglia and their polarization. However, we provide some examples of compounds that have been tested for their ability to influence polarization toward an anti-inflammatory phenotype in multiple sclerosis models.

The first example is spermidine, a polyamine found in most organisms. It is biosynthesized from putrescine, which is obtained in the body from the amino-acid ornithine. Spermidine has been shown in vitro to inhibit pro-inflammatory cytokine secretion by LPS-stimulated microglia. It inhibits the expression of iNOS and COX-2, two major pro-inflammatory enzymes and decreases the secretion of NO, PGE_2_, IL-6, and TNF-α at the transcriptional level. These effects seem to be mediated through the inhibition of the NF-κB, PI3K/Akt, and MAPK signaling pathways [54]. Spermidine has also been tested for effects in vivo with the EAE model. After treatment with spermidine, the mice showed decreased EAE clinical scores and the severity of the disease was diminished in both preventive and curative treatment schemes. These repercussions are mediated through macrophages because spermidine is able to inhibit their pro-inflammatory polarization by diminishing the secretion of the pro-inflammatory cytokines IL-1β, IL-6, IL-12, and TNF-α and by decreasing their antigen-presentation to lymphocyte capacity. As a consequence, less T-cells are activated and the EAE severity is weaker [55,56]. To our knowledge, no clinical trial has been conducted to assess the effect of spermidine intake on multiple sclerosis. 

Resveratrol is a stilbene found in berries, grapes, and nuts. It is one of the most studied natural compounds in multiple sclerosis. Most of the studies agree on the protecting effect of resveratrol in the EAE model. Its administration reduces the severity of the disease in mice and favors remyelination. The activity is mediated through the inhibition of secretion of pro-inflammatory cytokines by macrophages (e.g., IL-6 and TNF-α), but also by promoting the activation of regulatory T cells (TH17) [57,58,59,60]. Nevertheless, one study by Sato et al. showed surprising results because resveratrol was responsible for an exacerbation of EAE disease in mice [61]. These counterintuitive results show that it is difficult to work with natural compounds as plants display an important variability in their composition and have a wide scope of different targets, thus the determination of the real effect is challenging. 

Another well studied compound is curcumin, a polyphenol extracted from the rhizomes of *Curcuma longa* L. This compound has been shown to ameliorate EAE. There are various mechanisms of action implicated in this neuroprotective effect, but curcumin also regulates microglia activation through limiting its secretion of pro-inflammatory cytokines (TNF-α, IL-6, IL-12) as well as monocyte activation by inhibiting their infiltration in the CNS [62,63,64,65,66]. Interestingly, two clinical trials have been conducted to assess the effect of curcumin on MS patients (NCT03150966 and NCT01514370). One shows a neuroprotective effect of curcumin by limiting the inflammation of the CNS in MS patients [67].

Forskolin found in the Indian Coleus, *Plectranthus barbatus* Andrews, is often used as a food supplement or traditional oriental medicine. The effect of forskolin has been tested on the EAE model. The substance presents an anti-inflammatory action on extracted macrophages and microglia: M2-like markers are upregulated (e.g., Arg1 and Mrc1) whereas M1-like markers are downregulated (e.g., CD86 and MHC class II). These effects are mediated through the activation of cAMP and the ERK signaling pathway. Consequently, the proliferation of autoimmune CD4 T cells decreases and finally the neuroinflammation caused by the disease decreases and the recovery phase is promoted [68].

## 4. Macrophages/Microglia in GBM

### 4.1. TAMs and M1-M2 Balance

Glioblastoma are the most frequent type of gliomas, but also one of the most aggressive. They are characterized by a very important mitotic activity, high vascular proliferation, and necrosis. Glioblastomas are highly invasive and infiltrate the surrounding tissues, although they do not metastasize [4]. These tumors are also characterized by their high capacity to evade the immune system [69]. Macrophages and microglia have a leading role in this immune evasion capacity and represent the majority of non-neoplastic cells of the tumor microenvironment. These macrophages and microglia possess a specific M2-like polarization and are called tumor associated macrophages or TAMs [5,8,70]. TAMs are in favor of GBM tumorigenesis because they do not support cytotoxic T cell activation and even inhibit their proliferation, as they secrete anti-inflammatory cytokines (e.g., IL-10 and TGF-β) and growth factors (e.g., EGF), thus promoting development and angiogenesis of the tumor [5,69,71,72]. This system is a vicious circle because TAMs are promoted by the tumor itself, which secretes various factors such as IL-6, IL-10, TGF-β, and PGE2, and in return, TAMs promote the tumor’s proliferation and survival [12,69]. A solution to break this circle would be to inhibit the anti-inflammatory phenotype of TAMs and promote a repolarization toward an inflammatory phenotype [72]. 

### 4.2. Treatments Influencing Polarization for GBM Care

Targeting TAMs in glioblastoma is an interesting alternative, knowing the fact that they represent up to 30% of the cells constituting the tumor bulk [5]. Contrary to that of MS, no treatment on the market for GBM is currently targeting macrophage/microglia polarization. Thus, some compounds studied for their effect in this field will be reviewed here. These compounds are either small molecules, antibodies, or nucleotides that target five major signaling pathways: CSF1 and its receptor, CD47, CD40, TLRs, and STAT. 

#### 4.2.1. Colony Stimulating Factor 1 and Its Receptor

The first and most studied target is colony stimulating factor 1 (CSF1) and its receptor (CSF1R, also called CD115). CSF1R is a tyrosine-kinase receptor activated by the binding of the ligand CSF1, followed by homodimerization of the receptor. This signaling pathway is implicated in the differentiation of monocytes into M2-like, anti-inflammatory macrophages. Thus, inhibiting the activation of the receptor by targeting the ligand or the receptor is an interesting opportunity [73,74,75]. An important number of compounds were designed, tested in vitro, in vivo, and even clinically. All of these are small molecules or antibodies [74]. The most studied among them is a small molecule inhibitor called pexidartinib (or PLX3397), which can be taken orally and is able to cross the BBB. It showed interesting results in vitro and in mice by blocking glioblastoma progression, proliferation, and evasion [76,77]. Unfortunately, a clinical trial conducted on 37 patients with recurrent glioblastomas showed no effect of the molecule. However, pexidartinib has a good safety profile, is well tolerated by the patients, and is able to reach the tumor tissue [78]. The lack of effects of the treatment alone does not mean that it is not reaching the target or influencing it. An interesting option is to assess the effects of pexidartinib in combination with other known treatments. An enhancement of the efficacy has been shown in vivo on tumor bearing mice by combining PLX3397 and other antiangiogenic tyrosine-kinase inhibitors (dovitinib or vatalanib). An inhibition of tumor cell proliferation and an increase in tumor-cell apoptosis as well as a downregulation of M2 markers in TAMs compared to PLX3397 given alone has been observed [76]. A clinical trial has been conducted to assess the efficacy of the combination of pexidartinib, radiotherapy, and temozolomide (NCT01790503), but the results are not available at this moment [79].

Another well studied molecule is BLZ945, which is also a brain penetrating tyrosine-kinase inhibitor with a strong affinity for CSF1R. This compound is able to inhibit growth, progression, and survival of glioblastoma tumor cells in vivo by “re-educating” the TAMs and increasing phagocytosis of tumor cells. In fact, BLZ945 induces a downregulation of M2 markers in TAMs rather than depleting them [80]. However, Quail and colleagues showed that although BLZ945 strongly induces tumor regression in mice, after a few weeks of treatment, a resistance to the compound appeared and the tumor rebounded. This resistance is mediated through the TAMs. They compensate CSF1R inhibition by upregulating the IGF-1/IGF-1R axis and the PI3K signaling pathway, which are in favor of tumor growth and malignancy. A M2-like pro-tumorigenic phenotype is restored via the upregulation of M2 markers (TGF-β, IL4 and CD206). To avoid this resistance, combinational therapy is needed by combining BLZ645 with inhibitors of the resistance pathways (e.g., IGF-1R inhibitors) [81]. Further studies are needed to assess the effects of BLZ945 in association with other compounds and for the moment, no clinical trial has been conducted for its effect on GBM.

#### 4.2.2. CD47

Another possibility is to target CD47 and inhibit its activation. CD47 is an integrin also called the “don’t eat me” signal found on tumor cells, which inhibits their phagocytosis by macrophages after the binding of CD47 and SIRPα (signal-regulatory protein α) found on macrophages. Inhibition of this axis leads to the induction of phagocytosis of the tumor cells by M1-like and M2-like macrophages. This induction is much more important in M1-like macrophages. Furthermore, the inhibition of the axis also leads to the re-polarization of TAMs from a M2-like to a M1-like phenotype. Inhibition of this axis is obtained by the use of anti-CD47 antibodies. Hu5F9-G4, a monoclonal humanized antibody showed anti-tumor effects on murine and human glioblastoma cell lines and in vivo on grafted mice [82]. Another antibody targeting CD47 has been used after surgical resection of GBM in rats and extended survival of the animals and retarded relapse of the tumor has been observed [83]. A last anti-CD47 antibody has been tested on glioma stem cells: it is also effective to induce phagocytosis of these cells in vitro and in vivo, thereby limiting tumor growth [84].

#### 4.2.3. CD40

The next target is CD40, a costimulatory molecule, member of the tumor-necrosis factor receptor family that is expressed on the surface of immune cells (B cells, dendritic cells, and macrophages), non-immune cells (endothelial and epithelial cells), and on tumor cells. Its activation by the ligand CD40L by activated T-cells, platelets, and macrophages induces the proliferation and differentiation of B cells and macrophages, which is accompanied by the promotion of antigen presentation and anti-tumor immunity [85]. Activating CD40 is a potential mechanism for targeting macrophage polarization, as shown for pancreatic adenocarcinoma. In a murine model of this cancer, a monoclonal anti-CD40 agonist antibody is able to induce tumor regression by repolarizing macrophages toward a pro-inflammatory phenotype [86]. Knowing the fact that in glioblastoma, upregulation of the CD40/CD40L axis is an indicator for better prognosis, it is interesting that in vivo, the use of this anti-CD40 agonist antibody actually shows anti-tumor effects [85,87]. This effect is mediated by pushing the macrophages toward a M1-like phenotype [88]. 

#### 4.2.4. Toll-like Receptors (TLRs)

Toll-like receptors (TLRs) are also a studied target. This family of receptors is part of the pattern recognition receptors (PRRs) involved in the initiation of innate immune response by recognizing microbial-associated molecular patterns (MAMPs) and danger-associated molecular patterns (DAMPs) present on microorganisms. Their activation induces cytokine secretion, opsonization, phagocytosis, activation of the complement system, and proliferation [89]. Although there is a discrepancy regarding the role of TLRs in tumors, some agonists have been tested for their effects on glioblastoma. The most studied among them are phosphorothioate oligodeoxynucleotides (ODN) containing unmethylated cytosine-guanosine motifs (CpG) and targeting TLR9. These immunostimulatory substances are able to induce tumor death and to increase the survival of grafted rats and mice through an effect on the immune system. Indeed, in immunodeficient mice, the effect of ODN is abrogated [90,91]. CpG-ODNs were also appraised for their effects on humans. In two different phase I clinical trials, it was shown in patients with recurrent GBM that CpG-ODNs are generally well tolerated, except in some cases of lymphopenia, mild fever, seizures, and transient neurological worsening. It was also observed that the survival of patients was slightly extended compared to patients treated with temozolomide [92,93]. However, in a phase II trial, CpG-ODNs injected directly in the brain tumors of patients with recurrent GBM only showed modest activity. The radiological responses were low, but the number of long-term survivors was higher than in other studies. This example shows an important limit in the use of compounds targeting macrophages and the immune system in general: the considerable patient- and tumor-dependent response to the compounds [94]. Another limitation of these compounds as well as other TLR agonists (e.g., imidazoquinolines) is the absence of specificity, which can be the source of side effects. As a matter of fact, their activity is not only mediated through macrophages, but also through NK cells, dendritic cells (DCs), and T cells [89,91,95]. 

#### 4.2.5. Signal Transducers and Activators of Transcription (STATs)

Signal transducers and activators of transcription (STATs) also serve as targets for the repolarization of macrophages/microglia. STAT3 is implicated in the M2-polarization process and its inhibition is an interesting option to favor a pro-inflammatory and anti-tumoral state. Such an inhibitor is the small molecule WP1066. This compound is able to activate macrophages/microglia and to drive them toward a pro-inflammatory state. Results in vitro and in vivo showed that WP1066 is able to reduce glioblastoma viability and growth [12,96,97]. Its efficacy is mediated through the induction of the expression of costimulatory molecules (CD80 and CD86) by macrophages and microglia, the secretion of pro-inflammatory cytokines (IL-2, IL-12, and IL-15), and the induction of the proliferation of effector T-cells [98]. Two phase I clinical trials to study WP1066 and its effect on gliomas including glioblastomas are currently recruiting (NCT01904123 and NCT04334863).

#### 4.2.6. Other Targets

Aside these well studied targets and compounds, many others constitute some promising routes for investigation. Among them, we can cite the target CXCR4 inhibition by a cyclic peptide inducing a switch of macrophages toward a M1-like phenotype associated with a reduction in the proliferation and dissemination of human GBM cells in vitro and in vivo [99]. Another compound we can cite is amphotericin B, an antifungal medicine, which is able to enhance the capacity of macrophages and microglia to impede brain tumor-initiating cell proliferation and survival. A prolonged survival was then observed in grafted mice [100]. A last interesting mechanism is the use of a bispecific neutralizing antibody (vanucizumab) targeting angiopoietin-2 and VEGF. The antibody is able to reprogram macrophages and microglia toward an anti-tumoral phenotype, thereby delaying GBM growth and prolonging survival of grafted mice. Interestingly, using the antibody directed against both targets showed better effects than targeting VEGF alone (which is the mechanism of action of the FDA-approved drug bevacizumab for GBM). Additionally, knowing the fact that vanucizumab is safe for use in humans (NCT01688206 clinical trial), it constitutes an interesting compound for further studies [101].

All of the above cited compounds target M2-like macrophages and microglia to induce a repolarization toward a M1-like state. However, another mechanism has also been studied: inhibiting the recruitment of TAMs with an anti-inflammatory phenotype. One example is cyclosporin A, which permits limiting the infiltration of these cells, but its lack of selectivity is responsible for important side effects including the development of tumors [102]. A more selective option is to use an antibody directed against one specific target. One example is an anti-CCL2 (or anti-MCP1) antibody that is able to extend modestly, albeit significantly, the survival of GBM tumor-bearing mice in which a diminution in the number of TAMs in the tumor microenvironment has been observed [103]. 

## 5. Conclusions

Multiple sclerosis and glioblastoma are two very different diseases. However, in both cases, the immune system has an important role. In MS, an auto-immune mechanism leads to an inflammatory reaction directed against one’s own body and is responsible for demyelination and neurodegeneration [6]. In GBM, local inhibition of the immune system and a tolerance driven by the tumor itself drives the tumor survival and proliferation [5]. Targeting the immune system is an interesting choice. In the first pathology, the exacerbated immune response should be regulated whereas in the second pathology, it should be re-activated against the tumor cells. Critical contributors of the immune system are the cells from the monocyte lineage, particularly macrophages and microglia. Both are implicated in the pathophysiology of MS and GBM due to their capacity to endorse different phenotypes of activation. On one hand, their classical mode of activation or M1-like phenotype has a pro-inflammatory profile and it has been shown that this phenotype is prevalent in active MS lesions [3]. On the other hand, their alternative mode of activation or M2-like phenotype with an anti-inflammatory profile is predominant in cancerous diseases [70]. Knowing these facts, we wanted to review the treatments for both effects on the market or under development, which can specifically influence the polarization of macrophages and/or microglia, in order to re-educate them. For MS (Figure 4), we showed that most of the immunomodulating treatments target these cells by inhibiting their pro-inflammatory attributes and promoting a repolarization toward an anti-inflammatory phenotype. Other synthetic or natural occurring compounds are also under development with the purpose to re-educate macrophages and microglia. 

For GBM (Figure 5), small molecules, antibodies, and nucleotides are under development with the aim to re-activate macrophages and microglia, thereby favoring anti-tumoral immunity. For the moment, no compound has been approved on the market with a mode of action specifically targeting the polarization of macrophages/microglia.

However, considering the results of the studies reviewed here, we believe that it is a promising approach deserving more attention and research. New methods could be used to alter macrophage/microglia polarization. An example of an original method that showed promising results in models for both pathologies is the use of membrane targeting peptides (MTPs). For GBM, two MTPs have been developed: one targeting plexin-A1 and another targeting neuropilin-1 [104,105]. For MS, the peptide targeting plexin-A1 has been shown to promote remyelination [106]. These compounds are able to modulate the activation of a specific receptor by binding specifically to its transmembrane (TM) domain. They can either inhibit the activation of the receptor by interfering with the correct oligomerization or favor its activation by inducing conformational changes [107,108]. Knowing the fact that some receptors such as CSF1R are implicated in the mechanism of polarization, therefore, it is conceivable to design such peptides to block or improve these receptors and thereby favor one or the other polarization.

## Figures and Tables

**Figure 1 pharmaceutics-14-00344-f001:**
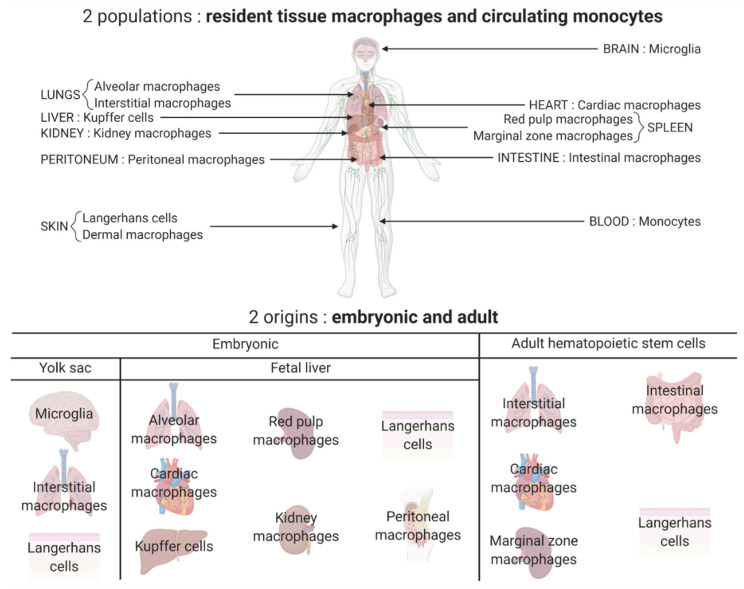
Different populations and origins of macrophages. Two major populations of macrophages coexist in the body: the macrophages that reside and self-renew in the tissues and the ones differentiated from the monocytes circulating in the blood and originating from the hematopoietic stem cells. Most tissue resident macrophages have an embryonic origin and arise either from the yolk sac (for example microglia in the brain) or directly from the fetal liver (for example, Kupffer cells in the liver and kidney macrophages). Some tissue resident macrophages are peripherally derived such as intestinal macrophages. All these tissue resident macrophages constitute a heterogeneous population with tissue-specific functions.

**Figure 2 pharmaceutics-14-00344-f002:**
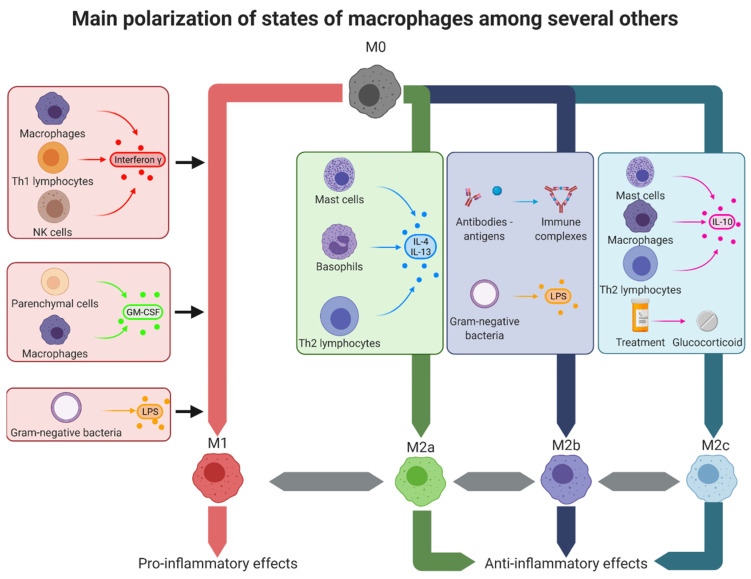
Macrophage polarization states. In the tissues, macrophages can adopt several polarization types corresponding to different activation states. The theoretical extremes of these states are, on one hand, the M1-like macrophages with pro-inflammatory effects, and on the other hand, the M2-like macrophages with anti-inflammatory effects. M1-like macrophages can be obtained by stimulation with interferon-γ, secreted by other M1-like macrophages, Th1 lymphocytes, and NK cells; GM-CSF secreted by M1-like macrophages and parenchymal cells or LPS present on Gram-negative bacteria. M2a alternatively activated macrophages were obtained by stimulation with IL-4 and IL-13 secreted by mast cells, basophils, and Th2 lymphocytes. M2b type II macrophages were obtained after stimulation with immune complexes or LPS. Finally, M2c deactivated macrophages were obtained by stimulation with IL-10 secreted by mast cells, M2-like macrophages, and Th2 lymphocytes or after treatment with glucocorticoids such as dexamethasone. GM-CSF: granulocyte-macrophage colony-stimulating factor, IL: interleukin, LPS: lipopolysaccharide, Th: T helper.

**Figure 3 pharmaceutics-14-00344-f003:**
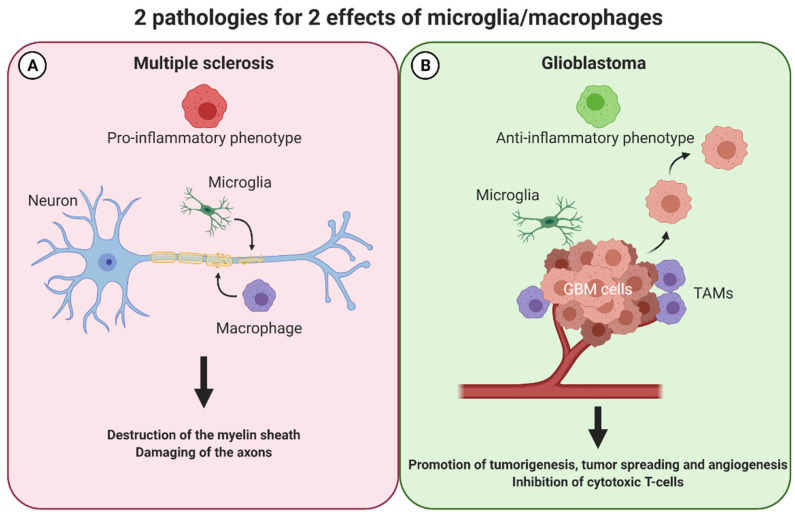
Macrophages have opposing effects in two pathologies of the central nervous system: multiple sclerosis and glioblastoma. In multiple sclerosis (**A**), macrophages exert a M1-like phenotype and thereby attack and destroy the myelin sheath and damage the axons in the CNS. In glioblastoma (**B**), these cells adopt the opposite M2-like phenotype favoring an anti-inflammatory milieu. Consequently, tumorigenesis, tumor-spreading, and angiogenesis is promoted whereas cytotoxic T-cells are inhibited. CNS: central nervous system, TAMs: tumor associated macrophages.

**Figure 4 pharmaceutics-14-00344-f004:**
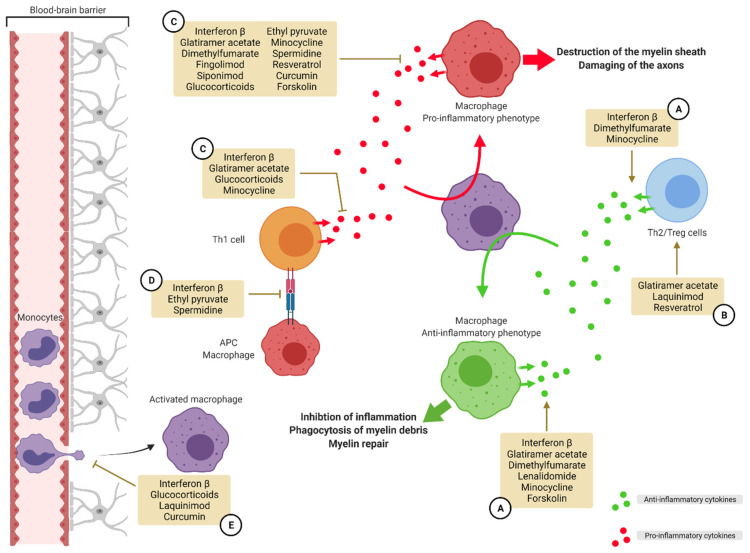
Targeting macrophage polarization in multiple sclerosis. In MS, macrophages adopt a M1-like phenotype promoting inflammation. A strategy for the resolution of inflammation and myelin sheath destruction is to target this polarization and push the macrophages toward a M2-like anti-inflammatory phenotype. Several compounds on the market and in development use different methods to re-educate macrophages. One strategy is to promote the secretion of anti-inflammatory cytokines by macrophages and Th2/Treg lymphocytes (**A**). Another strategy is to promote the activation of Th2 and Treg lymphocytes (**B**). Another way to achieve this purpose is to inhibit the secretion of pro-inflammatory cytokines by macrophages and Th1 lymphocytes (**C**). Preventing antigen presentation by macrophages is another strategy (**D**). A last strategy is to limit extravasation of monocytes into the CNS (**E**). APC: antigen presenting cell, MS: multiple sclerosis, Th: T helper, Treg: regulatory T cells.

**Figure 5 pharmaceutics-14-00344-f005:**
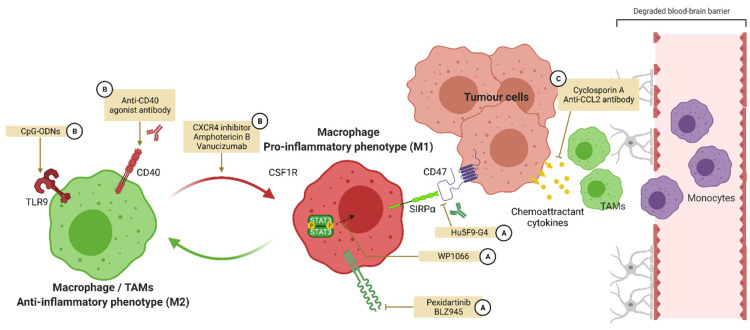
Targeting macrophage polarization in glioblastoma. In GBM, macrophages adopt a M2-like phenotype inhibiting inflammation and anti-tumoral immunity. A strategy for the reactivation of this immunity is to target the polarization of macrophages in order to promote M1 activation. A first strategy is to inhibit targets promoting M2 polarization such as CSF1R and STAT3 or the recognition of CD47 by SIRPα (**A**). Another strategy is to activate targets favoring M1 polarization such as TLR9 and CD40 (**B**). A last strategy is to inhibit the attraction of M2 polarized TAMs from the circulation toward the CNS (**C**). CSF1R: colony-stimulating factor 1 receptor, SIRPα: signal regulatory protein α, TAMs: tumor associated macrophages, TLR: Toll-like receptor.

## Data Availability

Not applicable.

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
