# Peer review of "Manipulating Macrophage/Microglia Polarization to Treat Glioblastoma or Multiple Sclerosis"

_pharmaceutics, 2022, doi:10.3390/pharmaceutics14020344_

Round 1

Reviewer 1 Report

In this article, the authors focus on different polarization states of macrophage/microglia in different pathologies such as multiple sclerosis and glioblastoma and review promising treatment strategies targeting macrophage/microglia polarization in these pathologies. Although this therapeutic approach could be complicated and contain a broad spectrum of applications, Figure 4 and 5 are quite simple and helpful to understand the therapeutic agents on the market, in preclinical or clinical development.

This review paper is well-written and serve as a reference to the readers.

In Page 7 Line 224, “one” should be omitted.

Author Response

Corrections were done along recommendations

Reviewer 2 Report

In this manuscript, Kuntzel and Bagnard reviewed therapeutic opportunities in the treatment of glioblastoma and multiple sclerosis by manipulating the polarization of macrophage and microglia. The topic is interesting and should be of interest to a broad research area. The manuscript is well written with balanced discussion of each disease.

Author Response

thank you for excellent comment

Reviewer 3 Report

1 The review is very well written, just need to have flow or continuation on the subject.

2. Need to rearrange few sentence like " The different polarization of states of macrophages" here you have to describe about the detail about the polarization in basic to standard.

Author Response

  • Transition words have been added to improve the reading flow
  • Sentences have been corrected